# Retinoblastoma Survival Following Primary Enucleation by AJCC Staging

**DOI:** 10.3390/cancers13246240

**Published:** 2021-12-13

**Authors:** Junyang Zhao, Zhaoxun Feng, Gareth Leung, Brenda L. Gallie

**Affiliations:** 1Department of Pediatric Oncology Center, Beijing Children’s Hospital, Capital Medical University, National Center for Children’s Health, Beijing 100045, China; zhaojunyang@163.com; 2Department of Ophthalmology, University of Ottawa, Ottawa, ON K1L 8L6, Canada; zfeng@toh.ca (Z.F.); gleun065@uottawa.ca (G.L.); 3Department of Ophthalmology and Vision Science, Hospital for Sick Children, Toronto, ON M5G 1X8, Canada; 4Krembil Research Institute, University Health Network, Toronto, ON M5T 2S8, Canada; 5Techna Institutes, University Health Network, Toronto, ON M5T 2S8, Canada; 6Departments of Ophthalmology and Vision Science, Molecular Genetics, and Medical Biophysics, University of Toronto, Toronto, ON M5T 3A9, Canada

**Keywords:** retinoblastoma, enucleation, survival, AJCC, classification, staging, prognosis

## Abstract

**Simple Summary:**

Despite the advent of new eye salvage therapies for retinoblastoma in recent years, upfront eye removal remains a life-saving treatment modality. We retrospectively evaluated the AJCC 8th edition cancer staging for prediction of survival of 700 patients consecutively managed with primary enucleation. Overall, 5-year survival was 95.5% and 5-year disease-specific survival was 95.7%. Patients with enucleation <26 days from diagnosis had better survival than those delayed >26 days. The eyes of children with raised intraocular pressure with neovascularization and/or buphthalmos (cT3c) had worse survival than children without this feature (cT2b, cT3b, and cT3d). Children with evidence of extraocular tumor on pathology (pT4) had dramatically worse survival than those with intraocular tumor (pT1 to pT3d). Survival was better for children with high-risk pathology (pT3/pT4) eyes after six cycles of adjuvant chemotherapy than those who underwent fewer cycles.

**Abstract:**

Primary enucleation of the eye with retinoblastoma is a widely accessible, life-saving treatment for retinoblastoma. This study evaluated the survival of patients following primary enucleation based on AJCC 8th edition staging. Included were 700 consecutive patients (700 eyes) treated with primary enucleation at 29 Chinese treatment centers between 2006 and 2015. Excluded were patients with less than one year follow-up, bilateral retinoblastoma, clinical evidence of extraocular disease at diagnosis, or prior focal or systemic therapy. The 5-year overall survival was 95.5%, and 5-year disease-specific survival (DSS) was 95.7%. Survival was better when enucleation was <26 days from diagnosis than delayed >26 days (96.1% vs. 86.9%; *p* = 0.017). Patients with eyes presenting with raised intraocular pressure with neovascularization and/or buphthalmos (cT3c) had worse 5-year DSS (87.1%) than those without (cT2b, 99.1%; cT3b, 98.7%; cT3d, 97.2%) (*p* < 0.05). The 5-year DSS based on pathological staging was pT1 (99.5%), pT2a (95.5%), pT3a (100%), pT3b (93.0%), pT3c/d (92.3%), and pT4 (40.9%). Patients with pT3 pathology who received six cycles of adjuvant chemotherapy had better 5-year DSS (97.7%) than those with no chemotherapy (88.1%; *p* = 0.06) and those who underwent 1–3 cycles (86.9%, *p* = 0.02) or 4–5 cycles (89.3%, *p* = 0.06). Patients with pT4 pathology who received six cycles of chemotherapy had better 5-year DSS than those with 0–5 cycles (63.6% vs. 16.7%; *p* = 0.02). Prompt primary enucleation yielded high long-term survival for children with retinoblastoma. The AJCC 8th edition staging is predictive of survival.

## 1. Introduction

Retinoblastoma, the most common primary malignant intraocular tumor in children, affects 1 in 16,000 to 18,000 live births [1]. Untreated, intraocular retinoblastoma is universally fatal with orbit, brain, and distant metastases [2]. The primary goal of retinoblastoma treatment is saving life, while secondary goals are preservation of eye and vision without compromising safety.

Enucleation, or removal of the affected eye, was the only treatment for retinoblastoma from the mid-1800s until external beam radiation therapy was developed in the mid-1900s [3]. The utility of primary enucleation has diminished with the advent of eye salvage therapies, including systemic chemotherapy [4,5,6,7], intra-arterial chemotherapy [8,9], intravitreal chemotherapy [10,11], and pars plana vitrectomy [12]. However, primary enucleation remains the safest treatment for intraocular retinoblastoma, especially for advanced intraocular disease in resource-limited settings [13,14]. Large retrospective studies have shown that delay to enucleation by prolonged attempted eye salvage is associated with diminished survival [15,16].

When chemotherapy became the predominant eye salvage modality in 1996 [4,5,6,7], three slightly different, similarly named eye staging systems were published to predict successful globe salvage: International Intraocular Retinoblastoma Classification (IIRC) (Murphree, Children’s Hospital of Los Angeles 2005) [17], International Classification for Retinoblastoma (ICRB) (Shields, Will’s Eye Hospital Philadelphia 2006) [18], and the Children’s Oncology Group (COG 2011) [19]. A 2017 survey of staging in 39 global retinoblastoma treatment sites indicated that 33% of centers used COG, 27% Murphree IIRC, and 23% Shields ICRB, while 17% were uncertain which version was used [20]. This lack of uniform staging creates barriers to objective comparisons of patient outcomes.

The American Joint Committee on Cancer (AJCC) Cancer Tumor, Node, Metastases Staging (TNM) retinoblastoma staging was developed by the Ophthalmic Oncology Task Force (7th edition 2009) [21] and revised in the 2017 8th edition [22] to optimize a common language for clinical care, research, and pathology for ocular cancers, including retinoblastoma. Retinoblastoma is the first cancer to include hereditary (H) in the TNM classification to distinguish germline retinoblastoma with risks of bilateral disease and second cancers.

An international, multicenter, collaborative study of metastasis-related survival of 2190 patients stratified based on AJCC 8th edition staging showed that both clinical staging and pathological staging prognosticate for survival [23]. However, a limitation of that study was the lack of detailed treatment information and heterogeneity of treatment protocols between centers. Moreover, pre-enucleation chemotherapy has now been shown to decrease the frequency of high-risk histopathology features but not actually lower survival, suggesting that interpretation of pathological staging is unreliable [15,24]. We now report a cohort of patients managed with primary enucleation by one Chinese retinoblastoma team, eliminating the confounding effects of treatment variation and pre-enucleation chemotherapy. Our objective is to assess the prognostic value of AJCC 8th edition staging on survival in a cohort uniformly treated with primary enucleation.

## 2. Materials and Methods

### 2.1. Data Collection and Ethics

This is a multicenter retrospective study of all patients who received primary enucleation for intraocular retinoblastoma at 29 Chinese treatment centers between 2006 and 2015. The final follow-up date was 26 December 2020. All patients with retinoblastoma in this study were managed by one retinoblastoma team led by J.Z. (retinoblastoma specialist) under uniform protocol. J.Z. routinely traveled to the 29 hospitals and was the physician responsible for most of the retinoblastoma patients managed at these centers. Patients often traveled between centers to follow-up with J.Z. based on his clinic schedule. This care model aims to minimize long-distance travel for children and their families.

Data collected included sex, laterality, age at diagnosis, clinical staging, pathological staging, adjuvant chemotherapy and dates of diagnosis, enucleation, last follow-up, and death. The IIRC staging and AJCC 7th edition pTNM were recorded at the time of clinical encounters and were retrospectively converted to cTNM staging and AJCC 8th edition pTNM by J.Z. and Z.F. via review of clinical records, clinical photos, pathological reports, and representative microscopic slides. Due to missing documentation on the degree of scleral invasion (invasion of the inner 1/3 vs. outer > 2/3), pT3c cannot be distinguished from pT3d, and they were analyzed as one combined group (cT3c/d) in this study. All examinations under anesthesia and ophthalmic treatment were done by the same ophthalmology team from Beijing. Institutional Ethics Committee approval was obtained at Beijing Children’s Hospital in accordance with the Declaration of Helsinki.

### 2.2. Eligibility

All patients were screened for extraocular disease at diagnosis by magnetic resonance imaging (MRI) or computed tomography (CT) of orbit. For patients with hereditary or bilateral retinoblastoma, MRI or CT of the brain was also performed. MRI was the imaging of choice. CT was performed in centers without access to MRI or for families who elected CT due to financial restraint. Exclusion criteria included follow-up time less than 1 year (unless death occurred within 1 year), clinical or imaging findings of extraocular disease (orbital or systemic) at diagnosis, bilateral retinoblastoma, and children who received any focal or systemic treatment prior to enucleation.

### 2.3. Treatment

The systemic chemotherapy regimen for adjuvant treatment was intravenous carboplatin 560 mg/m^2^ on day 1, etoposide 150 mg/m^2^ on days 1 and 2, or teniposide 230 mg/m^2^ on day 2 and vincristine 1.5 mg/m^2^ on day 2 with 28 days/cycle. For patients with pT4 pathology, sometimes a higher dose of chemotherapy was used. However, this was individualized for each child at the discretion of the medical oncology team. There was variation in the number of chemotherapy cycles due to a multitude of factors, including parental choice (financial constraint, concern about side-effects, or perception that the disease was cured), variation in medical oncology protocols at different institutions, and medical contraindications to chemotherapy (severe thrombocytopenia, neutropenia, or adverse event from chemotherapy side effects).

External beam radiotherapy (EBRT) was offered as adjuvant treatment to patients with pT4 pathology, in addition to systemic chemotherapy. However, only a few selected centers in China perform EBRT for children and the cost of treatment is high. Furthermore, parents were concerned about the risk of secondary malignancy. For these reasons, few pT4 patients in our cohort had EBRT.

### 2.4. Statistical Analysis

Continuous and categorical variables were described using medians/range and frequency/percentage, respectively. Receiver-operating-characteristic analysis was used to define the threshold to categorize children into groups based on time from diagnosis to enucleation. Kaplan –Meier was used to estimate overall survival (OS) and disease-specific survival (DSS), with log-rank test to compare survival between groups. Disease-specific survival is defined as death due to retinoblastoma metastasis. Length of survival was measured from time of diagnosis to death. Patients who were alive were censored at last follow-up. All reported *p* values are two-sided, and less than 0.05 indicated significance. All analyses were performed using SPSS Version 25 (IBM Corp, Armonk, NY, United States).

## 3. Results

### 3.1. Clinical Characteristics and Treatments

A total of 700 patients (700 eyes) met the study criteria with a median follow-up of 92.3 months (range, 0.9–172.7 months). Baseline characteristics of patients are presented in Table 1. The first presenting signs were leukocoria (467, 67%), red eye (91, 13%), strabismus (85, 12%), low vision (25, 4%), epiphora (9, 1%), exophthalmos (7, 1%), and incidental findings on eye screening (16, 2%). Of the studied patients, three (0.4%) had a family history of retinoblastoma.

All 700 patients were treated with primary enucleation for unilateral retinoblastoma. The median time from diagnosis to enucleation was 0 months, or same-day enucleation (range, 0–22.0 months). Following enucleation, 318 (45%) patients received adjuvant chemotherapy with median six cycles (range, 1–15 cycles). In total, 122/436 (28%) patients with low-risk pathology (pT1/pT2) received adjuvant chemotherapy, and 175/194 (90%) patients with high-risk pathology (pT3/pT4) received adjuvant chemotherapy. Some parents refused adjuvant chemotherapy for children with high-risk pathology in enucleated eyes. Adjuvant EBRT was performed in 3/22 (14%) patients with pT4 pathology. Of 194 patients with high-risk pathology, 160 (82.5%) had follow-up longer than 3 years.

### 3.2. Survival

In total, 31 patients died, with 30 deaths due to retinoblastoma metastasis. The median time from diagnosis to metastatic death was 11.2 months (range, 4.3–37.9 months). One child died from sepsis 0.9 months following enucleation. The 5-year overall survival (OS) was 95.5% (95% CI: 93.9%–97.1%), and 5-year disease-specific survival (DSS) was 95.7% (95% CI: 94.1%–97.2%). Median time from diagnosis to metastatic death was 11.2 months (range, 4.3–37.9 months).

Receiver-operating-characteristic analysis identified the time from diagnosis to enucleation > 0.85 months (26 days) as the appropriate threshold for analysis of DSS. Survival was better for patients enucleated <26 days from diagnosis than >26 days (5-year DSS of 96.1% vs. 86.9%; *p* = 0.017).

Patients with IIRC Group E eyes had worse survival than those with Group D eyes (5-year DSS of 93.7% vs. 99.1%; *p* < 0.001) (Table 2). According to AJCC cTNM 8th edition [22], patients with cT3c had 5-year DSS of 87.1%, significantly worse than those with cT2b (99.1%), cT3b (98.7%), or cT3d (97.2%) (*p* < 0.05) (Table 2, Figure 1).

There was no significant survival difference between pT2a (100%), pT2b (95.5%), pT3a (100%), pT3b (93.0%), and pT3c/d (92.3%) (*p* > 0.05). Patients with pT4 eyes had the worst 5-year DSS (40.9%, *p* < 0.05; Table 3, Figure 2).

By pT3 pathology staging, patients treated with six cycles of adjuvant chemotherapy had better DSS (Figure 3A) than those with 1–3 cycles (5-year DSS of 97.7% vs. 86.9%; *p* = 0.021), but not different than those with no chemotherapy (5-year DSS of 97.7% vs. 88.1%; *p* = 0.065) or 4–5 cycles of chemotherapy (5-year DSS of 97.7% vs. 89.3%; *p* = 0.061). There was no difference in DSS between patients with six cycles vs. seven or more cycles (5-year DSS of 97.7% vs. 100%; *p* = 0.629).

Adjuvant chemotherapy for patients with pT4 pathology who were treated with six cycles had better DSS than those treated with 0–5 cycles (5-year DSS of 63.6% vs. 16.7%; *p* = 0.017) (Figure 3B). There was no difference in DSS after adjuvant chemotherapy for pT4 patients treated with six cycles or >six cycles (5-year DSS of 63.6% vs. 20.0%; *p* = 0.256).

### 3.3. Analysis of Patients Who Died

Of 405 patients with low-risk histopathology eyes, three (two pT1 and one pT2b) died from metastatic disease. All had timely same-day enucleation of cT3c eyes (raised intraocular pressure with neovascularization and/or buphthalmos) and received no adjuvant chemotherapy.

pT3 high-risk histopathology was found in 172 patients. Of those, 11 died from metastatic disease (10 pT3b, 1 pT3d), one patient had 8 months’ delay from diagnosis to enucleation, and the parents of two refused adjuvant chemotherapy. Patients with only massive choroidal invasion had 5-year DSS of 100% compared to 95.4% for those with only retrolaminar optic nerve head invasion not involving the transected optic nerve end (excluding cases with both features) (*p* = 0.389). Patients with tumor invasion of the transected end of the optic nerve had worse survival than those with retrolaminar optic nerve head invasion not involving the transected end (5-year DSS of 36.8% vs. 95.4%; *p* < 0.001). Of 22 patients with pT4 histopathology, 13 died of metastatic disease; 2/13 had significant delay from diagnosis to enucleation (11.7 and 18.1 months) and 1/13 received no adjuvant chemotherapy due to parental refusal.

The median time from last cycle of adjuvant chemotherapy to death was median 6.0 months (range, 0.5–18.8 months). Only one patient with pT4 pathology had time between last chemotherapy to death < 2 months. This patient had four cycles of adjuvant chemotherapy, but additional treatment was prematurely stopped because of death.

## 4. Discussion

Enucleation of the diseased eye to prevent tumor metastasis is commonly used for advanced intraocular disease with poor potential for vision, especially when the other eye is normal. While enucleation is life-saving, parental refusal to allow removal of the dangerous eye remains a leading cause of death in developing countries [2]. In the retinoblastoma centers reporting in this study, the most common indication for primary enucleation was advanced retinoblastoma with poor visual prognosis and a functional contralateral eye. We retrospectively reviewed the survival outcome of 700 unilateral patients managed with primary enucleation without influence of a second diseased eye or pre-enucleation chemotherapy and assessed accuracy of the AJCC retinoblastoma staging [22] to predict survival of patients with intraocular retinoblastoma.

In 700 consecutive patients treated with primary enucleation, we observed 5-year OS of 95.5% and 5-year DSS of 95.7%. There were two cohorts in the United States treated with primary enucleation that had higher OS: 98% 5-year Lu et al. [25] and 98% 2-year Yannuzzi et al. [26]. In the AHOPCA II multicenter study of Central American patients with unilateral retinoblastoma (International Retinoblastoma Staging System Stage I; eye enucleated, completely resected histologically), the 102 patients who received primary enucleation had 5-year OS of 94% [16]. In low- and middle-income countries, treatment abandonment occurs for a variety of reasons, including financial constraints, residing long distances from treatment centers, and the false perception that the disease was cured [27]. We suspect the more common treatment abandonment in the Chinese and Central American settings may account for the slightly lower survival.

Patients with primary enucleation <26 days from diagnosis had better survival than those with a longer delay. The common reasons for delayed enucleation were parental hesitancy for enucleation and children with symptomatic infections which precluded safe anesthesia. In certain cases, parents initially abandoned treatment but returned to care after some time, leading to prolonged delays between diagnosis and enucleation. In our multicenter study of patients receiving pre-enucleation chemotherapy, delay between diagnosis and enucleation >3.5 months for Murphy IIRC Group D eyes and >2 months for Group E eyes decreased survival [15]. Our present study shows that, in the absence of pre-enucleation chemotherapy, the window of opportunity to maximize survival is narrower.

IIRC and AJCC clinical staging were designed to prognosticate the likelihood of successful eye preservation with chemotherapy. However, clinical staging can also prognosticate patient survival. In an international multicenter study of 2085 patients, AJCC clinical staging correlated with higher mortality [28]. Our cohort primarily focused on advanced intraocular disease (cT2b, cT3b, cT3c, and cT3d) in primarily enucleated children. We found that patients with cT2b (tumors with vitreous seeding and/or sub-retinal seeding) eyes had comparable 5-year survival to cT3b (tumor invasion of the pars plana, ciliary body, lens, zonules, iris, or anterior chamber), 99.1% vs. 98.7%, consistent with the increasing recognition that anterior chamber invasion is not an independent risk factor for metastasis [29]. The AJCC pathological staging groups’ tumor invasion of iris stroma, trabecular meshwork, or Schlemm’s canal as pT2b, not considered a high-risk feature.

We found that cT3c (raised intraocular pressure with neovascularization and/or buphthalmos) had worse survival than cT3d (hyphema and/or massive vitreous hemorrhage) (5-year DSS of 87.1% vs. 97.2%) (Table 2) or any lower clinical staging, suggesting a revision for the 9th edition of AJCC. We hypothesize that large posterior tumors and weaker sclera may predispose cT3c eyes to metastasis through transscleral or optic nerve tumor extension. In contrast, cT3d eyes are generally not considered for eye salvage because media opacity precludes safe monitoring, while metastatic risk may not be significantly associated with intraocular bleeding.

Of 405 patients with low-risk histopathology (pT1b and pT2b), three (0.7%) died, all with cT3c clinical stage eyes. Compared to cT3d, cT3c is not associated with longer delay to enucleation or greater proportion of high-risk histopathology (pT3/pT4; 40% vs. 42%; *p* = 0.664). These findings suggest that raised IOP with neovascularization and buphthalmos may be an independent risk factor for metastasis.

The AJCC retinoblastoma classification is designed to predict children at risk of metastasis and guide post-enucleation treatment. Focal choroidal invasion *or* pre-/intra-laminar optic nerve invasion is pT1. Only eyes with concomitant presence of both features are staged pT2a. Both pT1 and pT2a in our cohort showed very high 5-year survival of 99.5% and 100%, respectively. Anterior chamber tumor invasion is staged pT2b with 5-year OS of 95.5%; this represents 1 of the 22 patients who died.

The 5-year OS was 100% for children with pT3a eyes (massive choroidal invasion > 3 mm in largest diameter, or multiple foci of focal choroidal invasion totaling > 3 mm, or any full-thickness choroidal involvement). All 16 pT3a patients in our study were treated with adjuvant chemotherapy, indicating that pT3a pathology confers high survival with adjuvant chemotherapy. A retrospective study by Bosaleh et al. demonstrated that eyes with massive choroidal invasion had a higher chance of orbital/extraocular relapse compared to those with focal choroidal invasion (5.8% vs. 0.8%), but OS was not different (98.7% vs. 99.2%) [30]. It has been postulated that massive choroidal invasion, by itself, is not a high-risk feature. Future studies with large sample size are warranted to explore the necessity of adjuvant chemotherapy to maintain high survival of patients with pT3a eyes.

The 5-year survival for pT3b (retrolaminar invasion of the optic nerve head) and pT3c/d (partial or full-thickness invasion of the sclera) eyes was 93.0% and 92.3%, respectively. These eyes show a trend to worse survival compared to lower pathology staged eyes but do not demonstrate statistical significance. Finally, patients with pT4 extraocular disease (transscleral invasion or tumor at the transected end of the optic nerve) showed dramatically diminished 5-year survival of 40.9%. pT4 pathology suggests a positive surgical margin with residual unresected tumor; therefore, the likelihood of disease progression and metastasis is markedly higher than in lower pathology stages.

Even with high-resolution magnetic resonance imaging (MRI), the gold-standard imaging modality for retinoblastoma, the sensitivity to detect high-risk features is limited. Hiasat et al. showed that MRI sensitivity to detect post-laminar invasion and extra-scleral extension was 42% and 67%, respectively [31]. Thus, diagnostic imaging alone cannot reliably rule out high-risk pathological features. This highlights the importance of prompt enucleation in children with high clinical suspicion of advanced intraocular retinoblastoma [32,33].

Finally, we showed that the 5-year survival of patients with pT3 and pT4 pathology is optimized by six cycles of adjuvant chemotherapy at 97.7% and 63.6%, respectively. Fewer cycles of chemotherapy yielded worse survival, while more cycles did not further improve survival. Using six cycles of adjuvant chemotherapy is common practice at many retinoblastoma centers around the world [7,34,35,36]. To our knowledge, this is the first study to demonstrate that this protocol yields superior survival for patients with pT3 and pT4 pathology.

The major strengths of this study are its large sample size, inclusion of only unilateral patients, management by one retinoblastoma team, uniform treatment of primary enucleation, and extended follow-up. This enabled study of survival outcome stratified by AJCC staging without interference from other treatment modalities or disease processes in the contralateral eye. A limitation of this study is its retrospective nature. AJCC 8th edition was not available at the time of data collection; therefore, the eyes were re-staged based on review of clinical records, pathological reports, and representative microscopic slides. In addition, pathology staging was performed at multiple centers across the country. As such, the accuracy of staging is influenced by the level of experience of the pathologist.

## 5. Conclusions

To our knowledge, this is the first study to evaluate the AJCC 8th edition in a dataset restricted to unilateral retinoblastoma patients receiving primary enucleation. We showed that prompt primary enucleation yields good long-term survival. Following primary enucleation, the 5-year OS was 95.5% and 5-year DSS was 95.7%. We demonstrated that timely primary enucleation (<26 days) optimizes survival. We showed that clinical staging, traditionally used to predict likely success of eye salvage, is also predictive of long-term overall survival. We demonstrated that cT3c (raised intraocular pressure with neovascularization and/or buphthalmos) predicts worse survival, independent of pathology staging. We confirmed that higher pathological staging generally correlates with worse survival, with pT4 pathology associated with 40.9% 5-year DSS. This study supports the common practice of six cycles of adjuvant chemotherapy to maximize survival for eyes with high-risk histopathology (pT3/pT4). The AJCC 8th edition staging is a common language with which clinicians can prognosticate outcome, tailor treatment, and communicate findings through research.

## Figures and Tables

**Figure 1 cancers-13-06240-f001:**
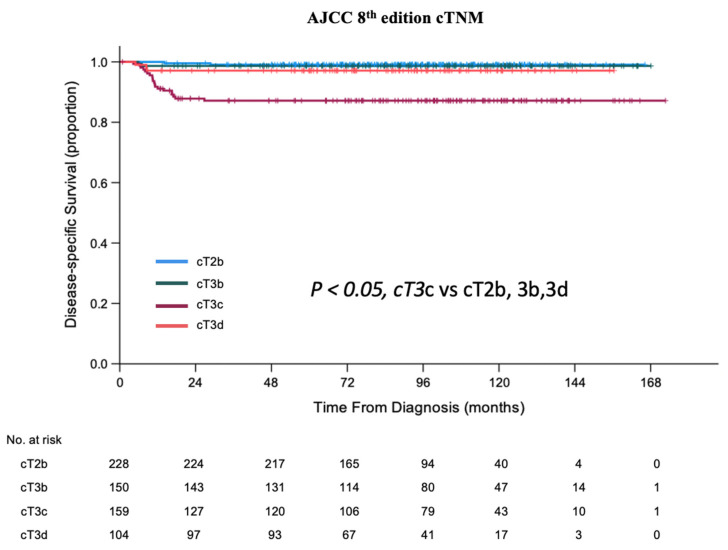
Kaplan–Meier curves of disease-specific survival (DSS) of patients treated with primary enucleation classified by AJCC 8th edition cTNM.

**Figure 2 cancers-13-06240-f002:**
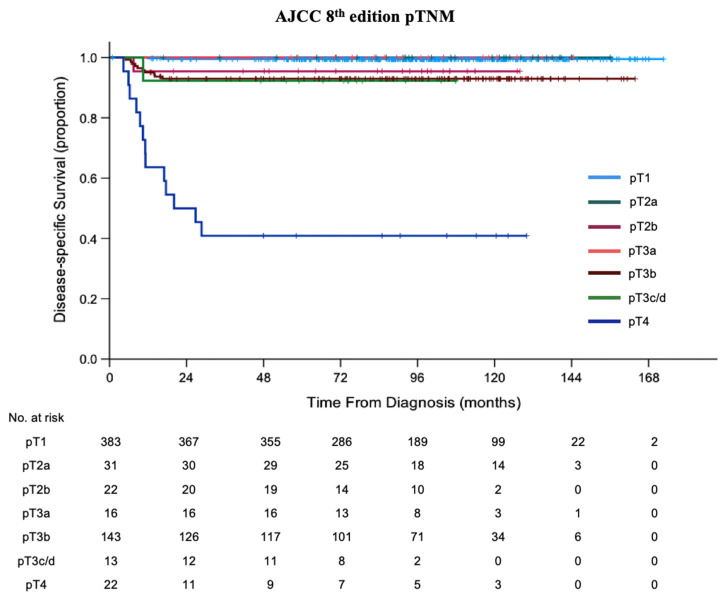
Kaplan–Meier curves of disease-specific survival (DSS) of patients treated with primary enucleation classified by AJCC 8th edition pTNM.

**Figure 3 cancers-13-06240-f003:**
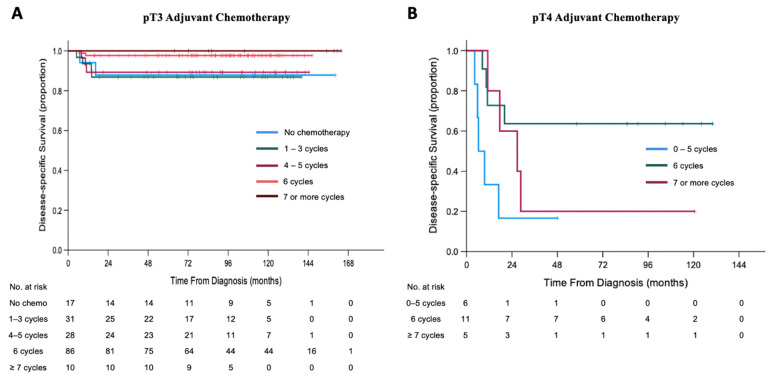
Kaplan–Meier curves of disease-specific survival (DSS) of patients treated with different cycles of adjuvant chemotherapy for AJCC 8th edition. (**A**) pT3 pathology. (**B**) pT4 pathology.

**Table 1 cancers-13-06240-t001:** Baseline clinical characteristics.

Characteristic		Total Patients *n* = 700	Death *n* = 31
Sex	Male	404 (58%)	19 (61%)
	Female	296 (42%)	12 (39%)
Age of diagnosis (months)	Median	24	29
	Range	1–212	4–63
Laterality	Right	357 (51%)	17 (55%)
	Left	342 (49%)	14 (45%)
Clinical Staging AJCC 8th edition			
cT2b	Vitreous and/or subretinal seeding	228 (32%)	2 (6%)
cT3a	Phthisis or pre-phthisis bulbi	3 (0.4%)	0 (0%)
cT3b	Tumor invasion of choroid, pars plana, ciliary body, lens, zonules, iris, or anterior chamber	150 (22%)	2 (6%)
cT3c	Raised intraocular pressure with neovascularization and/or buphthalmos	159 (23%)	21 (68%)
cT3d	Hyphema and/or massive vitreous hemorrhage	104 (15%)	3 (10%)
cT3e	Aseptic orbital cellulitis	2 (0.3%)	1 (3%)
Unknown		54 (8%)	2 (6%)
Pathological Staging AJCC 8th edition			
pT1	Intraocular tumor(s) without any local invasion, focal choroidal invasion, or pre- or intralaminar involvement of the optic nerve head	383 (55%)	3 (10%)
pT2a	Concomitant focal choroidal invasion and pre- or intralaminar involvement of the optic nerve head	31 (5%)	0 (0%)
pT2b	Tumor invasion of stroma of iris and/or trabecular meshwork and/or Schlemm’s canal	22 (3%)	1 (3%)
pT3a	Massive choroidal invasion (>3 mm in largest diameter, or multiple foci of focal choroidal involvement totaling >3 mm, or any full-thickness choroidal involvement)	16 (2%)	0 (0%)
pT3b	Retrolaminar invasion of the optic nerve head, not involving the transected end of the optic nerve	143 (20%)	10 (32%)
pT3c/d	Any partial-thickness involvement of inner two thirds sclera; full-thickness invasion into outer third of sclera and/or invasion into or around emissary channels	13 (2%)	1 (3%)
pT4	Evidence of extraocular tumor: tumor at transected end of optic nerve, tumor in meningeal spaces around optic nerve, full- thickness invasion of sclera with invasion of episclera, adjacent adipose tissue, extraocular muscle, bone, conjunctiva, or eyelids	22 (3%)	13 (42%)
Unknown		70 (10%)	3 (10%)

**Table 2 cancers-13-06240-t002:** Kaplan-Meier Disease-specific Survival Pairwise Comparison for AJCC Clinical Staging.

Classification	Variable	5-Year Kaplan–Meier DSS Estimate (95% CI), %		
IIRC Classification (*n* = 646)	Group D (*n* = 228)	99.1 (97.9–100)		
	Group E (*n* = 418)	93.7 (91.3–96.1)		
AJCC cTNM (*n* = 641)	cT2b (*n* = 228)	99.1 (97.9–100)		
	cT3b (*n* = 150)	98.7 (96.8–100)		
	cT3c (*n* = 159)	87.1 (81.8–92.7)		
	cT3d (*n* = 104)	97.2 (93.8–100)		
Pairwise Comparison *p*-value for AJCC		cT2b	cT3b	cT3c
cT2b				
cT3b		0.663		
cT3c		<0.001	<0.001	
cT3d		0.158	0.384	0.007
Overall Wilcoxon Log-Rank *p* < 0.001				

**Table 3 cancers-13-06240-t003:** Kaplan–Meier Disease-specific Survival Pairwise Comparison for AJCC Pathological Staging.

Classification	Variable	5-Year Kaplan–Meier DSS Estimate % (95% CI)					
AJCC 8th edition pTNM	pT1 (*n* = 383)		99.5 (98.7–100)				
	pT2a (*n* = 31)		100				
	pT2b (*n* = 22)		95.5 (86.6–100)				
	pT3a (*n* = 16)		100				
	pT3b (*n* = 143)		93.0 (88.7–97.3)				
	pT3c/d (*n* = 13)		92.3 (77.5–100)				
	pT4 (*n* = 22)		40.9 (20.0–61.9)				
Pairwise Comparison *p*-value		pT1	pT2a	pT2b	pT3a	pT3b	pT3c/d
pT1							
pT2a		0.685					
pT2b		0.031	0.235				
pT3a		0.771	-	0.394			
pT3b		<0.001	0.133	0.177	0.281		
pT3c/d		0.003	0.123	0.719	0.267	0.935	
pT4		<0.001	<0.001	<0.001	<0.001	<0.001	0.005
Overall Wilcoxon Log-Rank *p* < 0.001							

## Data Availability

Raw data are available in the Appendix A.

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
