# Peer review of "Retinoblastoma Survival Following Primary Enucleation by AJCC Staging"

_cancers, 2021, doi:10.3390/cancers13246240_

Round 1

Reviewer 1 Report

The authors are to be commended for an impressive cohort with large sample size. The study is timely as there is a need for more data on the Retinoblastoma survival by AJCC staging. 

I would suggest a few points for the authors to clarity before publication:

- under methods - extraocular disease was excluded, however under table pt4 disease (evidence of extraocular tumor) was included - they mentioned this is a multi Center study, yet said all cases were managed and treated by the same ophthalmology team? Did the same team review all clinical notes, pathological slides to reclassify according to TNM 8th edition? Maybe this can be clarified.  - not sure why ct3a cases were not included in table 2.  - there were cases included with long delay between diagnosis till primary enucleation (with no interim treatment in between), eg 22 months wonder if the reason behind this can be explained or explored?  - it would be interesting to know if all cases received preop MRI and whether the MRI features correlated or predicted the pathological features

Author Response

The authors are to be commended for an impressive cohort with large sample size. The study is timely as there is a need for more data on the Retinoblastoma survival by AJCC staging. 

We appreciate the reviewer’s interests in our work. We are also thankful of the thoughtful suggestions which help to improve the quality of our manuscript.

under methods - extraocular disease was excluded, however under table pt4 disease (evidence of extraocular tumor) was included 

This is a very good point. We now clarify in the method section to say, “clinical or imaging findings of extraocular disease (orbital or systemic) at diagnosis”. (Lines 113-114)

As such we are referring to radiological or gross clinical findings of extraocular disease. Patients with pT4 disease (pathological evidence of extraocular disease) are not excluded because pT4 can only be determined after enucleation. In fact, all the patients who died from tumor metastasis by definition have developed extraocular disease at some point. Our exclusion criteria focuses on the absence of extraocular features “at diagnosis” prior to enucleation of the eye. This is an important point to clarify and we thank the reviewer for bringing this to our attention.

they mentioned this is a multi Center study, yet said all cases were managed and treated by the same ophthalmology team

Thank you pointing out the unusual multicenter collaboration behind this study. J.Z is the most responsible physician for all retinoblastoma children managed at these centers. In reality, the patients are not tied to any particular center and often travel between centers for follow-up based on J.Z’s schedule.

We added in the method section, “All patients with retinoblastoma in this study were managed by one retinoblastoma team led by J.Z (retinoblastoma specialist) under uniform protocol. J.Z routinely travels to the 29 hospitals and was the most responsible physician for retinoblastoma patients managed at these centers. Patients often travel between centers to follow-up with J.Z based on his clinic schedule. This care model aims to minimize long-distance travel for children and their families.” (Lines 90-95)

Did the same team review all clinical notes, pathological slides to reclassify according to TNM 8th edition? Maybe this can be clarified.

Yes, two authors (J.Z and Z.X.F) reviewed the clinical notes, photos, pathological reports and representative pathology slides to restage from original staging to the AJCC 8th edition staging.

In the method section, we added “by J.Z and Z.X.F.” (line 100).

not sure why ct3a cases were not included in table 2

Table 2 outlines the 5-year survival estimate of patients by AJCC cTNM. As shown in Table 1, we only had 3 patients with cT3a. This small sample size is insufficient for survival estimate. Just because all three patients survived does not mean the subgroup has 100% survival. Therefore, cT3a is omitted in the sub-group analysis of survival based on AJCC cTNM. For similar reason, cT3e (n=2) is also omitted from the 5-year survival estimate.

there were cases included with long delay between diagnosis till primary enucleation (with no interim treatment in between), eg 22 months wonder if the reason behind this can be explained or explored? 

We now added to the discussion, “The common reasons for delayed enucleation were parental hesitancy for enucleation and children with symptomatic infections which precluded safe anesthesia. In certain cases, parents initially abandon treatment but returned to care after some time, leading to prolonged delay between diagnosis and enucleation.” (Lines 258-261)

The 22-month delay would be the third scenario, where parents initially abandoned treatment, then later regretted their decision and returned to care.

it would be interesting to know if all cases received preop MRI and whether the MRI features correlated or predicted the pathological features

We now clarify our prep-op screening for extraocular disease: “All patients were screened for extraocular disease at diagnosis by magnetic resonance imaging (MRI) or computed tomography (CT) of orbit. For patients with hereditary or bilateral retinoblastoma, MRI or CT of brain were also performed. MRI was the imaging of choice, CT was performed in centers without access to MRI, andfor families who elected CT due to financial restraint.” (Lines 109-114)

We agree that correlation between imaging and pathological features is an interesting topic. The absence of extraocular features on imaging but subsequent pT4 (extraocular disease) on pathology suggests a discordance, commented on in Discussion (Lines 327-333). However, full evaluation of radio-pathological correlation is outside the scope of this paper. We hope to revisit this topic in a future manuscript. For this type of study, it would be helpful to select patient screened with one imaging modality (MRI) and restrict to patients with short delay between imaging and enucleation.

Reviewer 2 Report

HI, this article investigates the retinoblastoma survival after enucleation by AJCC Staging. it provides the statistic view for guiding the people about enucleation and adjuvant chemotherapy. however, I think several question should be addressed in detail. 1. The Figure 1 should contain the MRI image of each stage. 2. Please discuss why you use a disease free or disease-specific model? 3. Please discuss the clinical impact since enucleation is still the major treatment of retinoblastoma. What 's important impact comes from this article? 4. What's the type of chemotherapy? 5. Whether the conclusion of good long-term survival were validated in other population? 6. Why pT4 has worse survival? Any difference between stages should be discussed in detail. 7. Author should complete the description of all the title, result and panel. For example, the title should be a short conclusion. the result should mention all the figure and panel. 3.

Author Response

The Figure 1 should contain the MRI image of each stage.

We thank the reviewer for taking an interest in our work and provide thoughtful suggestions to help improve the quality of this manuscript.

For our centers, MRI was used to exclude extraocular spread (orbital or systemic tumors) at the time of diagnosis. It is not clear to us how the MRI images could contribute to Fig 1. Clinical staging of these eyes was done based on clinical criteria which we outline in Table 1.

Please discuss why you use a disease free or disease-specific model? 

We thank the reviewer for raising this point. We now clarify in the methods, “disease-specific survival is defined as death due to retinoblastoma metastasis”. (Lines 137-138)

AJCC retinoblastoma staging is designed to predict outcome related to retinoblastoma disease progression. We think disease-specific survival is a more appropriate outcome measure as opposed to overall survival because it excludes death due to causes unrelated to retinoblastoma metastasis (e.g., infection, accident, other cancer). 

Please discuss the clinical impact since enucleation is still the major treatment of retinoblastoma. What 's important impact comes from this article?

We outlined the following major clinical findings of this manuscript in the Conclusion:

1.     Primary enucleation is a treatment modality that yields good long-term survival (5-year DSS 95.7%). (Lines 354-3355)

2.     Timely enucleation within < 26 days of diagnosis yield better survival than those with delay > 26 days (Lines 355- 356)

3.     Finding of raised intraocular pressure with neovascularization and/or buphthalmos (cT3c) predicts worst survival (Lines 358-360)

4.     6 cycles of adjuvant chemotherapy following enucleation maximized survival (Lines 361-363)

5.     AJCC 8th edition clinical and pathology staging can prognosticate survival (Lines 363-365)

These findings are directly applicable to clinical practice to improve outcome of children with retinoblastoma.

What's the type of chemotherapy?

This is a good point. We added the chemotherapy regimen to the method section, “The systemic chemotherapy regimen for adjuvant treatment was intravenous carboplatin 560 mg/m2 on day 1, etoposide 150 mg/m2 on days 1 and 2, or teniposide 230 mg/m2 on day 2 and vincristine 1.5 mg/m2 on day 2 with 28 days/cycle. For patients with pT4 pathology, sometimes a higher dose of chemotherapy was used. However, this is individualized for each child at the discretion of the medical oncology team.” (Lines 117-121)

Whether the conclusion of good long-term survival were validated in other population? 

We thank the reviewer for this suggestion. We now include in Discussion the following, “Two cohorts in the United States treated with primary enucleation had higher OS: 98 % 5-year Lu et al. [25] and 98% 2-year Yannuzzi et al. [26]. In the AHOPCA II multicenter study of Central American patients with unilateral retinoblastoma (International Retinoblastoma Staging System Stage I; eye enucleated, completely resected histologically), the 102 patients who received primary enucleation had 5-year OS of 94%.[16] In low- and middle-income countries, treatment abandonment occurs for a variety of reasons including financial constraints, long distance from treatment center and false perception that disease was cured [27]. We suspect the more common treatment abandonment in the Chinese and Central America settings, may account for the slightly lower survival.” (Lines 247-255)

Why pT4 has worse survival? Any difference between stages should be discussed in detail. 

We added the following clarification to explain the poor prognosis of pT4, “pT4 pathology suggests a positive surgical margin with residual unresected tumor, therefore the likelihood of disease progression and metastasis is markedly higher than lower pathology stages.” (Lines 324-326)

Author should complete the description of all the title, result and panel. For example, the title should be a short conclusion. the result should mention all the figure and panel

We thank the reviewer for this suggestion. However, as mentioned earlier, our paper generated multiple important conclusions, making it difficult to summarize all the findings in one statement. For this reason, we think our current open-ended title is appropriate. It captures the two main topic of the paper namely:

1.     Survival outcome following primary enucleation

2.     AJCC predictive value of survival

We have ensured that all our figures and tables are cited in the main text of the manuscript. 

Reviewer 3 Report

Lucid writing style.

The study objectively shows significance of current management protocols for unilateral retinoblastoma.

Author Response

We thank reviewer 3 for their interest and support of our work.

Reviewer 4 Report

In this report, Zhao et al. have conducted a retrospective analysis of overall and disease specific survival (DSS) of 700 children stratified by AJCC Staging after primary enucleation of unilateral retinoblastoma. The patients were treated at a 29 centers between 2006 and 2015. The grouping according to AJCC Staging 8th edition was performed retrospectively. The study shows a high overall survival of 95.5% for all children with primary enucleation. Clinical and pathological AJCC Staging correlated with disease specific survival of children with unilateral retinoblastoma. The authors conclude that AJCC 8th edition is predictive of prognosis and underlines the benefit of 6 cycles of  adjuvant chemotherapy.

The manuscript is well written and referenced. The number of figures and tables is appropriate for the information provided. There are some aspects listed below in which this manuscript could be improved.

Material and Methods:

  1. The authors do not mention radiotherapy and the reader can assume that radiotherapy was not part of the treatment regimen. Add the information whether any of the children received adjuvant EBRT or other type of radiotherapy in addition to the chemotherapy?
  2. Line 88: Clarify that and how one Chinese RB team (line 81) performed the treatment at 29 chinese centers (line 88).
  3. Exclusion criteria is- among other- extraocular disease . Specify that exclusion criteria are macroscopic extraocular disease because otherwise pT4 group patients have to be excluded.

Results:

Line 124: Explain, how the numbers add up: 318 received chemotherapy, while  122 were pT1/ pT2  and 175 were pT3/pT4. Which staging were the other 21 patients with adjuvant chemo or are these missing data?

Line 151: The group pT4 is broad and could potentially include children with metastatic disease. Exclusion criteria are: extraocular disease. Clarify that pT4 group only includes patients with microscopic extraocular disease and N0 M0 at diagnosis?

Table 3: The survival of 2 groups in table 3 is stated as 101 % while in reality highest possible survival is 100% even if statistical tests result in 101%. Correct to 100% for both groups.

Line 158: Specify the reasons for giving less than 6 cycles of chemotherapy .  if the chemotherapy was stopped prematurely because of relapse this causes a bias. If this information is not available then specify the number of patients that relapsed while being on chemotherapy compared to those that relapsed a reasonable time (at least 2 months) after a premature end of chemotherapy.

Line 167: see above: Are there pT4 patients that died/relapsed on chemotherapy and for this reason the chemotherapy was stopped prior to completing 6 cycles of chemotherapy?

Discussion:

Revise the conclusion if the reason for premature stop of chemotherapy has been relapse in a relevant number of patients (see comment above)

Author Response

The authors do not mention radiotherapy and the reader can assume that radiotherapy was not part of the treatment regimen. Add the information whether any of the children received adjuvant EBRT or other type of radiotherapy in addition to the chemotherapy?

We thank the reviewer’s thoughtful comments which help to improve the quality of our manuscript.

EBRT was not a mainstay adjuvant treatment in our cohort. Only few centers in China offer EBRT to children and it is associated with high cost ($50000-$60000 USD).

We added to the method section, “External beam radiotherapy (EBRT) was offered as adjuvant treatment to patients with pT4 pathology in additional to systemic chemotherapy. However, only few selected centers in China perform EBRT for children and the cost of treatment was high. Furthermore, parents were concerned about the risk of secondary malignancy. For these reasons, only few pT4 patients in our cohort had EBRT.” (Lines 127-133) We added to the results section, “Adjuvant EBRT was performed in 3/22 (14%) patients with pT4 pathology.” (Lines 171)

Line 88: Clarify that and how one Chinese RB team (line 81) performed the treatment at 29 chinese centers (line 88).

This is a very good question. J.Z is the most responsible physician for all retinoblastoma children managed at these centers. In reality, the patients are not tied to any particular center and often travel between centers for follow-up based on J.Z’s schedule.

We added in the method section, “All patients with retinoblastoma in this study were managed by one retinoblastoma team led by J.Z (retinoblastoma specialist) under uniform protocol. J.Z routinely travel to the 29 hospitals and was the most responsible physician for retinoblastoma patients managed at these centers. Patients often travel between centers to follow-up with J.Z based on his clinic schedule. This care model aims to minimize long-distance travel for children and their families.” (Lines 90-95)

Exclusion criteria is- among other- extraocular disease . Specify that exclusion criteria are macroscopic extraocular disease because otherwise pT4 group patients have to be excluded.

This is another good suggestion. We now clarify in the method section to say, “clinical or imaging findings of extraocular disease (orbital or systemic) at diagnosis”. (Lines 113-114)

As such we are referring to radiological or gross clinical finding of extraocular disease. Patient with pT4 disease (pathological evidence of extraocular disease) are not excluded. In fact, all the patient who died from tumor metastasis have developed extraocular disease at some point. Our exclusion criteria really focus on the absence of extraocular features “at diagnosis” prior to enucleation of the eye. We think this is an important point to clarify and we thank the reviewer for bring this to our attention.

Line 124: Explain, how the numbers add up: 318 received chemotherapy, while 122 were pT1/ pT2 and 175 were pT3/pT4. Which staging were the other 21 patients with adjuvant chemo or are these missing data?

The reviewer is correct that there are missing data. 70/700 (10%) eyes were enucleated at centers without dedicated ocular pathology support. The pathology report only noted whether or not the optic nerve margin was clear. Therefore, these eyes did not have enough information for pTNM staging. These patients are listed in Table 1 as “unknown” pTNM.

Line 151: The group pT4 is broad and could potentially include children with metastatic disease. Exclusion criteria are: extraocular disease. Clarify that pT4 group only includes patients with microscopic extraocular disease and N0 M0 at diagnosis?

This is very a very good point. We now clarify the exclusion criteria, “clinical or imaging findings of extraocular disease (orbital or systemic) at diagnosis” (Lines 113-114). The reviewer is correct that none of the children had overt orbital or metastatic disease “at diagnosis”. Macroscopic extraocular disease at diagnosis would have been excluded. But as mentioned in the discussion, the radio-pathological correlation is not always reliable for high-risk features.

Table 3: The survival of 2 groups in table 3 is stated as 101 % while in reality highest possible survival is 100% even if statistical tests result in 101%. Correct to 100% for both groups.

Thank you for pointing out this typo. It is now corrected.

Line 158: Specify the reasons for giving less than 6 cycles of chemotherapy .  if the chemotherapy was stopped prematurely because of relapse this causes a bias. If this information is not available then specify the number of patients that relapsed while being on chemotherapy compared to those that relapsed a reasonable time (at least 2 months) after a premature end of chemotherapy.

We thank the reviewer for raising this important point. We now outlined the reasons for variation in chemotherapy cycle, “There was variation in the number of chemotherapy cycles due to a multitude of factors including, parental choice (financial constraint, concern about side-effect or perception that disease is cured), variation in medical oncology protocols at different institutions and medical contraindications to chemotherapy (severe thrombocytopenia, neutropenia, or adverse event from chemotherapy side-effect).” (Lines 121-126)

Tumor relapse during adjuvant treatment was not one of the reasons for stopping chemotherapy prematurely. Patients with relapse would continue to receive chemotherapy usually at a higher dose. However, the reviewer points out an interesting possibility of ending chemotherapy prematurity because the patient is dead. We reviewed the data and now added the following information to Results: “The median time from last cycle of adjuvant chemotherapy to death was median 6.0 months (range, 0.5–18.8 months). Only one patient with pT4 pathology had time between last chemotherapy to death < 2 months. This patient had 4 cycles of adjuvant chemotherapy, but additional treatment was prematurely stopped because of death.” (Lines 23-234)

Line 167: see above: Are there pT4 patients that died/relapsed on chemotherapy and for this reason the chemotherapy was stopped prior to completing 6 cycles of chemotherapy?

We thank the reviewer for bring up this point. As discussed above, after review, we identified one patient with pT4 pathology whose adjuvant treatments was prematurely stopped because of death.

Revise the conclusion if the reason for premature stop of chemotherapy has been relapse in a relevant number of patients (see comment above)

Given there is only one patient whose adjuvant chemotherapy was prematurity terminated due to death, we think our conclusion of “6 cycles of chemo optimizing survival” is still appropriate.

Round 2

Reviewer 1 Report

The authors have provided response to all my suggestions satisfactorily. 

Reviewer 2 Report

Thank you for your response. 

Reviewer 4 Report

All comments have been addressed in detail. No further corrections.